# Naphthalimide-Piperazine Derivatives as Multifunctional “On” and “Off” Fluorescent Switches for pH, Hg^2+^ and Cu^2+^ Ions

**DOI:** 10.3390/molecules28031275

**Published:** 2023-01-28

**Authors:** Kristina Pršir, Mislav Matić, Marlena Grbić, Gerhard J. Mohr, Svjetlana Krištafor, Ivana Murković Steinberg

**Affiliations:** 1Department of General and Inorganic Chemistry, Faculty of Chemical Engineering and Technology, University of Zagreb, Marulićev trg 19, 10000 Zagreb, Croatia; 2Joanneum Research Forschungsgesellschaft mbH—Materials, Franz-Pichler-Straße 30, A-8160 Weiz, Austria

**Keywords:** 1,8-naphthalimide, fluorescent switch, pH, metal ion, heavy metals, chemosensors

## Abstract

Novel 1,8-naphthalimide-based fluorescent probes **NI-1** and **NI-2** were designed and screened for use as chemosensors for detection of heavy metal ions. Two moieties, methylpyridine (**NI-1**) and hydroxyphenyl (**NI-2**), were attached via piperazine at the C-4 position of the napthalimide core resulting in a notable effect on their spectroscopic properties. **NI-1** and **NI-2** are pH sensitive and show an increase in fluorescence intensity at around 525 nm (switch “on”) in the acidic environment, with p*K*_a_ values at 4.98 and 2.91, respectively. Amongst heavy metal ions only Cu^2+^ and Hg^2+^ had a significant effect on the spectroscopic properties. The fluorescence of **NI-1** is quenched in the presence of either Cu^2+^ or Hg^2+^ which is attributed to the formation of 1:1 metal-ligand complexes with binding constants of 3.6 × 10^5^ and 3.9 × 10^4^, respectively. The **NI-1** chemosensor can be used for the quantification of Cu^2+^ ions in sub-micromolar quantities, with a linear range from 250 nM to 4.0 μM and a detection limit of 1.5 × 10^−8^ M. The linear range for the determination of Hg^2+^ is from 2 μM to 10 μM, with a detection limit of 8.8 × 10^−8^ M. Conversely, **NI-2** behaves like a typical photoinduced electron transfer (PET) sensor for Hg^2+^ ions. Here, the formation of a complex with Hg^2+^ (binding constant 8.3 × 10^3^) turns the green fluorescence of **NI-2** into the “on” state. **NI-2** showed remarkable selectivity towards Hg^2+^ ions, allowing for determination of Hg^2+^ concentration over a linear range of 1.3 μM to 25 μM and a limit of detection of 4.1 × 10^−7^ M.

## 1. Introduction

Due to the well-known impact of heavy metals in the environment and on human health, the development of new, cost-effective, fast and reliable methods for their detection has become urgent [1,2]. Heavy metal detection methods have variously been proposed based on sensing mechanisms that include electrochemical and optical techniques with specifically designed recognition elements, such as molecularly imprinted polymers [3], nanomaterials [4], metal-organic frameworks [5] and N-heterocyclic receptors [6]. 

Fluorescence-based optical sensors and molecular probes in particular have attracted interest as metal ion sensors due to favorable properties such as high sensitivity and selectivity, the relatively simple detection process and intrinsic low cost. Various fluorescent chemosensors have been synthesized and investigated for heavy metal ion-determination based on photophysical mechanisms such as chelation-induced enhanced fluorescence (CHEF), chelation enhancement quenching (CHEQ), intramolecular charge transfer (ICT), photo-induced electron transfer (PET) and aggregation-induced emission (AIE) [7,8,9,10,11,12,13,14,15,16]. 

Photo-induced electron transfer sensors are particularly popular in molecular fluorescent “switch” designs. In PET systems, a fluorophore is covalently linked with a receptor through a short aliphatic spacer, while both segments maintain their individual properties. Upon interaction with the target analyte, the PET process is inhibited thereby causing changes in the quantum yield or fluorescence intensity [17,18,19]. 

The potent toxicity of mercuric ions to living systems has inspired the development of new fluorescent chemosensing molecules and sensing schemes for detection of Hg^2+^ in environmental, industrial and biological samples [2,20,21,22]. Chemosensors for Cu^2+^, a metal essential for life but also highly toxic to some organisms, have also been the subject of much research and development effort [22,23,24,25]. It is known that Cu^2+^ and Hg^2+^ act as quenching agents to many fluorescent probes. Cu^2+^ has intrinsic paramagnetic properties, which cause an energy or electron transfer, consequently decreasing the fluorescence intensity of the fluorophore [26]. Being a heavy atom, Hg^2+^ decreases the quantum yield of fluorescence by an external spin–orbital coupling effect, which induces a non-radiative intersystem crossing [27]. There is a great interest in developing probes for Hg^2+^ based on intensity enhancement, rather than quenching, since the latter often show lower sensitivity [28]. 

One notable group of fluorescent probes are the derivatives of 1,8-naphthalimide. Known for their chemical stability, these molecules exhibit excellent thermal, electrochemical, electroluminescent, optical and photophysical properties [29,30,31]. By introducing different functional moieties on the imide and amino nitrogen atom, it is possible to modify the physical and chemical properties of the naphthalimide. Such modifications then open the possibility of using the resulting molecules as chemical and biochemical sensors. In general, the molecular fragment bound to the 4-amino nitrogen forms the receptor part of the molecule that interacts with the analyte, while the fragment attached to the imide nitrogen atom affects the physical properties. 1,8-naphthalimide is a photostable conjugate system consisting of two aromatic rings and two double bonds with oxygen. This allows the π–π* electron transfer due to light absorption in the ultraviolet (UV) region [17]. In addition, naphthalimide derivatives exhibit large Stokes shifts. Thanks to these favorable photophysical properties and the relatively simple synthesis, naphthalimides have been used for both cell imaging and for measuring extracellular and intracellular pH. For example, naphthalimide-based pH-probes incorporating a protonable piperazine ring [32] and a hydroxyl group linked to the naphthalimide moiety at the C-4 position [33] have recently been reported. These demonstrate some of the many practical applications of naphthalimide derivatives as pH sensors in living systems for physiology and cell biology research. Naphthalimide probes and sensors for alkali [34], earth alkali [35] and heavy metal ions [36,37,38,39,40] have also been devised and investigated. Naphthalimides may also be used as components of functional materials [41] and as components of drug delivery systems [42], and are particularly suitable here due to the ease of structural modification by modulation of moieties attached to the amino nitrogen atom [43].

In this paper, we investigate the applicability of two compounds, **NI-1** and **NI-2**, as optical probes for pH and heavy metal ion detection (Figure 1). 

The **NI-1** and **NI-2** compounds each have a typical donor–spacer–acceptor structure (D-π-A), comprising a 1,8-naphthalimide core, which represents the fluorophore, and either a pyridin-2-ylmethyl or hydroxyphenyl moieties attached to the core through piperazine. The moieties act as additional receptor sites for metal ion binding. Due to the presence of the *N*-substituent containing a 2-hydroxyethylsulfonyl group, the covalent immobilization of these naphthalimide probes to different substrate materials and to nanoparticles (e.g., cellulose, textiles) is also possible using well-established vinyl sulfonyl immobilization chemistry. This molecular feature opens up a wide range of solid state applications for these probes in the future [44].

Structures of studied naphthalimide derivatives **NI-1** and **NI-2** are shown in Figure 1. The main structural difference between them is a molecular fragment attached via piperazine moiety at 2-hydroxyethylsulfonyl-functionalised naphthalimide. 

## 2. Results and Discussion 

### 2.1. Spectral Characteristics of ***NI-1*** and ***NI-2*** in Methanol

Structures of studied naphthalimide derivatives **NI-1** and **NI-2** are shown in Figure 1. The main structural difference between them is a molecular fragment attached via piperazine moiety at 2-hydroxyethylsulfonyl-functionalised naphthalimide. 

Absorption and emission spectra of compounds are shown in Figure 2a.

Both compounds displayed absorption spectra with the absorption maximum at 410 nm with molar absorption coefficients, *ε* = 2.41 × 10^4^ M^−1^ cm^–1^ for **NI-1** and *ε* = 8.52 × 10^3^ M^−1^ cm^−1^ for **NI-2**, which can be assigned to *π–π** transitions. To the naked eye, **NI-1** and **NI-2** displayed a very light yellow color (Figure 2b). Considering the fluorescence emission spectra, compounds **NI-1** and **NI-2** exhibited a single emission band with emission maxima at 523 and 525 nm, respectively, but the intensity was strongly dependent on the attached moiety. **NI-1**-bearing pyridine functionality is strongly green fluorescence emitting fluorophore, compared to hydroxyphenyl-substituted **NI-2**. Since the lone electron pair on the oxygen atom is a part of the conjugated system, overall electron density of the hydroxyphenyl moiety is increased, which might repulse the lone electron pair on the piperazine nitrogen. That in turn induces the electron transfer from the piperazine nitrogen to the fluorophore, which lowers the emission rate. On the other hand, the lone electron pair on the pyridine nitrogen is not involved in the conjugated system, making the ring overall electron-deficient. Compounds **NI-1** and **NI-2** exhibited large Stokes shifts in neutral pH (5558 cm^–1^ and 5629 cm^−1^) and therefore have great potential for the application as molecular probes, despite the observed low fluorescence quantum yield (*Φ*(**NI-1**) = 0.027, *Φ*(**NI-2**) = 0.004) [45]. 

### 2.2. Effect of pH on Spectral Properties 

The spectral properties of **NI-1** and **NI-2** as a function of pH were investigated using both absorption and fluorescence emission spectroscopy. The potential use of **NI-1** and **NI-2** as pH-sensing probes is based on the spectral changes caused by the protonation of the piperazine nitrogen atom. Optical pH sensors are characterized by their apparent p*K*_a_ values in buffer methanol mixtures (2:1), determined by spectrophotometry and spectrofluorimetry. The pH values correspond to the pH values of the aqueous buffers used.

#### 2.2.1. UV–Visible Absorption Spectra 

Changes in the pH of solutions caused minor changes in the absorbance spectra of the studied compounds (Appendix A). As the pH decreases from 7.0 to 1.6 the absorption band of **NI-1** gradually shifts from 410 nm to 388 nm, generally followed by an increase in absorbance of the maximum. The absorbance ratio at peak values (for protonated and neutral forms) has been used for calculating the apparent p*K*_a_ value (Appendix A). **NI-2** showed the same behavior upon the pH changes, with hypsochromic shift of absorption maxima from 410 nm to 389 nm upon the protonation along with the increase in the peak intensity, as well as a gradual hyperchromic shift in the pH range 5.0–8.0 (Appendix A). 

#### 2.2.2. Fluorescence Emission Spectra 

The emission spectral responses to pH for **NI-1** and **NI-2** and the dependence of fluorescence intensity (at 523 nm for **NI-1** and 525 nm for **NI-2**) upon the pH value are shown in Figure 3 and Appendix A. The fluorescence intensity changes of **NI-1** and **NI-2** upon pH were described well by a sigmoidal function (Boltzmann fitting), Figure 3b. 

The effect of pH is more pronounced in the fluorescence spectra than in the absorption spectra of compounds. It is known that the protonation of the nitrogen atom of piperazine stops the PET process from amine donor to the naphthalimide core and therefore, the fluorescence intensity increases gradually with the decrease in pH value [45,46,47]. The fluorescence intensity of both **NI-1** and **NI-2** were considerably increased upon the protonation, up to 29.6-fold for **NI-1** and 162.5-fold for **NI-2**. Fluorescence turn-ON in acidic solution supports the inhibition of the PET process in the piperazine–naphthalimide derivative upon the protonation of piperazine, as shown in Figure 4. 

The pH response curves exhibit excellent adjustment and apparent p*K*_a_ values were calculated to be 4.98 for **NI-1** (R^2^ = 0.9923) and 2.91 for **NI-2** (R^2^ = 0.9982). 

**NI-1** and **NI-2** have different substituents on their pH-sensitive amino group. The substituents did not affect the pH sensitivities, as both compounds displayed fluorescence enhancement upon the protonation. The different nature of substituent moieties attached to the protonable piperazine moiety (alkyl or aryl) significantly influenced the photophysical properties of derivatives, showing that the suitable substituents play a crucial role for tuning the p*K*_a_ and pH sensitivity. Moreover, **NI-1** and **NI-2** dyes were immobilized onto cellulose layers and the p*K*_a_ values of were determined to be 6.05 and 2.98, respectively [43]. These results indicate that both the nature of substituents and the immobilization technique have a significant impact on spectroscopic and pH sensing properties and could provide a different application of fluorescent naphthalimide dyes [48,49,50]. 

The properties of **NI-1** and **NI-2** in acidic (pH 1.6) and neutral solution (pH 7.0), including absorption and emission wavelength (*λ*_abs_, *λ*_emiss_), molar absorption coefficient (*ε*), fluorescence intensity, Stokes shift and calculated apparent p*K*_a_ values are summarized in Table 1. 

### 2.3. Metal Ion Sensing

The difference in molecular fragments attached via piperazine moiety at 2-hydroxyethylsulfonyl-functionalised naphthalimide, affected not only the emission spectra of **NI-1** and **NI-2**, but also their response upon binding with metal cations. Spectroscopic measurements of metal ion sensing were performed in MeOH/buffer (*v*/*v*, 1/2, pH 5.5). For the spectroscopic measurements, compound concentration and experimental conditions were adjusted for these two compounds separately, because of their different behavior in solution and response in the presence of metal cations. A pH value of 5.5 for the metal ion studies was chosen as a compromise to maximize sensitivity and to minimize effects of competing equilibrium reactions, i.e., protonation of **NI** at lower pH values and the metal ion hydrolysis at higher pH values.

#### 2.3.1. **NI-1**

Visual detection of metal ion sensitivity

To evaluate the metal ion sensitivity, preliminary tests were performed under a UV lamp (365 nm) by adding 30.0 μL metal ion stock solution to the compound in MeOH/buffer (*v*/*v*, 1/2, pH = 5.5) (*c* (**NI-1**) = 6.56 × 10^−6^ M) and the results are presented in Appendix A. 

**NI-1** exhibits bright green fluorescence in MeOH/buffer solution and with the addition of Cu^2+^, fluorescence quenching can be seen by naked eye. Addition of other metal ions did not cause any observable changes in the fluorescence of **NI-1**; 

Fluorescence titration experiments

The experiments were performed at compound concentration 5.00 × 10^−7^ M in MeOH/buffer (1/2, *v*/*v*, pH = 5.5) in the presence of 2, 20 and 95 equiv. of metal ions and the results are summarized in Figure 5. A small addition of metal ions (2 equiv.) did not significantly affect the fluorescence intensity of **NI-1**, while the presence of 20 equiv. of Cu^2+^ and Hg^2+^ resulted in fluorescence quenching of **NI-1** intensity for 46.4% and 22.7%, respectively. Furthermore, 95 equiv. of these two cations induced further quenching of fluorescence intensity of **NI-1** for 60.0% upon Cu^2+^ and 39.8% upon Hg^2+^ addition). Moreover, Al^3+^, Zn^2+^, Ba^2+^, Pb^2+^ and Cd^2+^ ions present in high concentrations (95 equiv.) induced **NI-1** fluorescence intensity increase in the range of 31.1 to 36.0% for Zn^2+^, Ba^2+^, Pb^2+^ and Cd^2+^ and 47.9% increase in the presence of Al^3+^, as summarized in Figure 5. 

Encouraged by these findings, we performed titration experiments with Cu^2+^ and Hg^2+^ and the results are presented in Figure 6 and Figure 7.

It can be seen from the emission spectra that an increase in the concentration of metal ions leads to a decrease in the fluorescence intensity, and a significant decrease in intensity is achieved at higher concentrations. In the case of Cu^2+^ ions, a hypochromic shift occurs, with the maximum of fluorescence intensity at 523 nm, while in the case of Hg^2+^ ions, a bathochromic shift occurs with a decrease in intensity. The wavelength of the fluorescence intensity maximum in the presence of Hg^2+^ ions shifts from 523 nm to 541 nm. Hypochromic shift is common in naphthalimide derivatives whose fluorescence is quenched in presence of Cu^2+^ ions [39]. However, bathochromic shift in the presence of Hg^2+^ ions is not common and the possibility of obtaining two different fluorescent responses can potentially be used to discriminate Cu^2+^ from Hg^2+^ using **NI-1**. 

Fluorescence properties of immobilized napthalimide derivatives with (methylpyridine)piperazine receptor in presence of Hg^2+^ and Cu^2+^ ions have been reported previously. Covalently immobilized probes on a glass carrier were examined at pH 7 in the presence of Hg^2+^ ions and have shown an increase in fluorescence with Hg^2+^ ions and a slight decrease with Cu^2+^ ions [37]. Additionally, Liu et al. [51] synthesized two naphthalimide derivatives, also with (methylpyridine)piperazine as receptor of **NI-1**, and immobilized them on (poly)styrene substrate. The resulting optodes differ only in the linker (*N*-substituent of naphthalimide) through which the compounds were attached to the carrier. In both cases, Hg^2+^ ions caused a significant increase in fluorescence intensity. It is interesting to notice that, although the receptor part of all four compounds is the same, their response to Hg^2+^ and Cu^2+^ ions differs significantly. This implies that other factors, such as immobilization method, material of the substrate, length of the linker part on the *N*-substituent, test conditions—particularly the pH of the solution, solvents and anion composition, etc., have a significant effect on the mechanism of the fluorescence response, possibly causing steric hindrance and influencing binding stoichiometry as well.

Unlike fluorescence emission spectra, UV-Vis absorption spectra of **NI-1** were not significantly affected by presence of Hg^2+^ and Cu^2+^ ions (Appendix A), showing an absorption maximum at 410 nm which can be assigned to the naphthalimide core; 

Reversibility tests

Reversibility tests were performed with the addition of EDTA to the solution containing **NI-1** and metal cations, in five cycles, as shown in Appendix A for Cu^2+^ and Hg^2+^, respectively. In general, upon addition of EDTA, the fluorescence of **NI-1** recovers. In the case of Cu^2+^, it was noticed that the recovered fluorescence intensity of **NI-1** increases by approx. 10% upon each addition of EDTA in repeating cycles. This behavior could be attributed to the enhancement of fluorescence of **NI-1** caused by other effects, possibly related to the gradually changing composition of the solution with each cycle. In comparison to free **NI-1**, the solutions containing **NI-1**, EDTA and excess Hg^2+^ ion show fluorescence intensity quenching, varying by approximately 8%, while the maximum shifts from 523 nm to 536 nm (Δ*λ*_max_ = 13 nm); 

Metal-binding studies

The stoichiometric coefficients of Cu^2+^ and Hg^2+^ ions upon binding with **NI-1** were assumed using Job’s plot as shown in Figure 8 and a 1:1 binding ratio is suggested for both ions, which is in accordance with previously published results for the same substituent [36,37].

The equilibrium association (binding) constant for Cu^2+^ and Hg^2+^ ions and **NI-1** binding was determined by Benesi–Hildebrand’s plot using the equation: (1)$$\frac{1}{F-F_{0}}=\frac{1}{K_{b}\left(-F_{0}\right){\left[Cu^{2+}\right]}^{n}}+\frac{1}{F_{max}-F_{0}}$$
where *F*_0_ is the fluorescence intensity in the absence of metal ions; *F*_max_—the fluorescence intensity in presence of saturated metal; *F*—the intensity of the metal-probe complex in the linear range and *n*—factor of stoichiometric binding (1 for 1:1 ratio). The correlation factor for both ions is R^2^ > 0.96 with the binding constant being 3.9 × 10^4^ for Hg^2+^ and 3.6 × 10^5^ for Cu^2+^ (Appendix A);

LOD (limit of detection)

The limit of detection (LOD) of **NI-1** for Cu^2+^ and Hg^2+^ ions was determined using the IUPAC-recommended equation: LOD = 3*σ*/*k*(2)
where *σ*_sp_ is the standard deviation of blank measurements and *k* is the slope between the fluorescence intensity and metal concentration in linear range as shown in Appendix A. The standard deviation for four blank measurements is *σ*_sp_ = 0.3295. The detection limit of **NI-1** is 1.5 × 10^−8^ M and 8.8 × 10^−8^ M for Cu^2+^ and Hg^2+^ ion, respectively. These values are comparable to similar fluorescent quenching chemosensors utilizing the naphthalimide structure with aliphatic amino receptors, reported previously [52,53,54,55]. Wu et al. [37] determined the limit of detection of Hg^2+^ ions by naphthalimide-based optode to be 2.00 × 10^−6^ M at pH 7.01 in 0.05 M Tris-HCl which is significantly higher than the detection limit of the compound **NI-1**. Furthermore, Gao et al. determined the detection limit of Cu^2+^ ions for three naphthalimide derivatives, which have the same receptor moiety as **NI-1**, in which fluorescence quenching occurs. The determined detection limits reported are in range from 2.62 × 10^−9^ to 1.17 × 10^−8^ M [39]. The Stern–Volmer constants for **NI-1** were calculated to be 152570 M^−1^ for Cu^2+^, and 23909 M^−1^ for Hg^2+^, as determined from the respective Stern–Volmer plots (Appendix A). Based on this evidence from the literature and the known behavior of Cu(II) as a strong paramagnetic quencher, we assume that the quenching here is also induced by chelation, rather than being collisional.

Table 2 shows a comparison between different copper sensors reported in the literature and the napthalimide sensor **NI-1** presented in this work.

#### 2.3.2. **NI-2**

Visual detection of metal ion sensitivity

**NI-2** did not fluoresce in the MeOH/buffer mixture at pH 5.5. Upon addition of metal cations, a drastic increase in the fluorescence intensity was observed only in the presence of Hg^2+^ ions (Figure 9). These initial experiments suggested that **NI-2** could be used as selective fluorimetric naked-eye chemosensor for Hg^2+^; 

Titration experiments

To investigate the possibility of an **NI-2** compound to detect Hg^2+^ ions, fluorescence titration was conducted. The fluorescence spectra were recorded at excitation wavelength of 410 nm. As shown in Figure 10, the maximum fluorescence band is centered at 525 nm. The spectrum of the **NI-2** compound without Hg^2+^ exhibits a low intensity peak around 533 nm, typical for the naphthalimide derivatives containing an amine substituent at the C-4 position of the naphthalene imide ring [61]. The addition of Hg^2+^ to the compound **NI-2** results in a remarkable enhancement of fluorescence signal but has nearly no effect on its UV-Vis spectrum (Appendix A). The fluorescence intensity increases steadily in response to the gradual increase in concentration of added Hg^2+^ ion in MeOH/buffer (*v*/*v*, 1/2, pH 5.5) at a compound concentration 6.53 × 10^–6^ M, while a hypsochromic shift Δ*λ_F_* = 6 nm was observed. Such a small shift suggests that upon addition of the metal ions the fluorophore ICT state is not affected and that the complexation takes place via a nitrogen atom on the piperazinyl part and hydroxyphenyl group of the substituent simultaneously, rather than the imide moiety. These results support the potential of the **NI-2** compound to be utilized as a highly selective and sensitive fluorescent sensor for Hg^2+^ ions; 

Selectivity

The selectivity towards Hg^2+^ ions was investigated by adding 150 µL of Hg^2+^ (1.00 × 10^−3^ M) to the **NI-2** solution (6.54 × 10^−6^ M) containing 150 µL of background metal ion (1.00 × 10^−3^ M) in MeOH/buffer mixture (*v*/*v*, 1/2, pH 5.5) (Figure 11). 

Compound **NI-2** shows a greater selectivity relative to **NI-1**, which can be attributed to the structural difference of their receptors. Namely, in **NI-2** the hydroxyphenyl group is attached directly to piperazine via a nitrogen atom which makes the receptor structure more rigid compared to **NI-1** in which pyridine is linked to piperazine by a methylene bond. The stiffer receptor structure in **NI-2** prevents it from coordinating various ions as is the case in **NI-1**;

Metal-binding studies

Using Job’s method, analogous to **NI-1**, the supposed stoichiometric binding coefficient of Hg^2+^ and **NI-2** was determined in MeOH/buffer mixture at pH 5.5. As shown in Appendix A, the dependence of fluorescence intensity at 525 nm on the molar content of **NI-2** in solution in the presence of Hg^2+^ ions might vaguely suggest a 2:1 metal–ligand binding ratio, which, given the structure of ligand, is however unlikely. We could conclude here that Job’s plot failed to reveal the real stoichiometry of the complex which is not unusual given the known limitations of the method [62]. 

The Benesi–Hildebrand’s method was employed by plotting the *F_0_*/(*F*–*F*_0_) against 1/[Hg^2+^], where *F* is the fluorescence intensity at 525 nm in the presence of the metal ion at the given concentration, and *F*_0_ is the fluorescence intensity of the free compound (Appendix A). The association constant was evaluated from the ratio of the intercept and the slope to be 8278 M^−1^, which is similar to previously reported results [37]; 

LOD (limit of detection)

As one of the most important sensor features for useful applications, the limit of detection (LOD) of **NI-2** for Hg^2+^ ions was determined using the IUPAC-recommended equation. 

Appendix A showed good linearity between the emission at 525 nm and concentrations of Hg^2+^ in the range from 0 to 25.0 μM, indicating that the compound **NI-2** can measure specific concentrations of Hg^2+^ quantitatively. 

In the linear range between 0–25 µM of the calibration curve, the limit of detection can be obtained as LOD = 3*σ*/k, where *σ* is standard deviation of blank measurements and *k* is the slope between the fluorescence intensity and the metal concentration. The standard deviation for four blank measurements is *σ* = 0,621 and the limit of detection was calculated to be 4.1 × 10^−7^ M, being comparable with similar chemosensors based on naphthalimides [26,37,63];

Reversibility

The reversibility test was performed by the addition of Hg^2+^ and EDTA to the **NI-2** solution. The addition of Hg^2+^ increased fluorescence intensity of **NI-2** as shown in Appendix A, but there was no change in the spectrum after the addition of EDTA to the solution of **NI-2**-Hg^2+^.

Table 3 shows a comparison between the different mercury sensors and napthalimide sensors **NI-1** and **NI-2** presented in this work.

It is important to note that nearly all the literature data in the area of chemosensors research for determination of complex stoichiometry, binding constants and quenching constants are based on the use of “old” graphical methods, such as Job’s plot, Benesi–Hildebrand (BH) or Stern–Volmer (SV) plots. Those methods show significant limitations and are often used non-critically to draw conclusions that cannot be made without more detailed analysis [67,68,69]. In this context, it is important to stress that all constants reported in our work are conditional constants. They were calculated under defined conditions, i.e., certain mixtures of solvents and pH values of buffer solutions. In addition to chelation equilibrium reactions assumed to be happening during metal-ion titrations, there are other possible reactions simultaneously happening in the solution including protolytic equilibrium of the sensor molecules and hydrolysis of metal cations. Calculation of exact binding constants would require more sophisticated calculation tools and different measurements, which for purely analytical applications is usually not necessary. Nevertheless, one has to be aware of the limitations of these graphical methods and use them accordingly. 

## 3. Materials and Methods 

### 3.1. Reagents and Instruments 

Indicator dyes **NI-1** and **NI-2** were from Joanneum Research Forschungsgesellschaft mbH (Austria) [43]. All chemicals and solvents were purchased from commercial suppliers Acros, Sigma Aldrich or Fluka and were used as received without further purification. Methanol for spectroscopic measurements was of spectroscopic (HPLC) grade. A phosphate buffer was used for the near neutral pH range (5.0–8.0) and citric buffer for the acidic pH range (1.2–4.8). Buffer solutions were prepared using chemicals of analytical grade following the procedure [70]. The pH values of buffer solutions were monitored using MA 5740 pH meter. 

The 1.0 × 10^−3^ M stock solutions of metal ions (Ag^+^, Mg^2+^, Ba^2+^, Cu^2+^, Hg^2+^, Zn^2+^, Co^2+^, Fe^2+^, Ni^2+^, Pb^2+^, Mn^2+^, Cd^2+^, Fe^3+^, Al^3+^) were prepared by dissolving metal salts (AgNO_3_, MgCl_2_, BaCl_2_, CuCl_2_, Hg(NO_3_)_2_, ZnSO_4_, Co(CH_3_CO_2_)_2_, FeSO_4_, NiSO_4_, Pb(NO_3_)_2_, MnCl_2_, (CH_3_CO_2_)_2_Cd, FeCl_3_, AlCl_3_) in 0.01 M HNO_3_. The UV-Vis spectra were recorded, against the solvent, at 25 °C, on a Varian Cary spectrophotometer in double-beam mode using quartz cuvette (1 cm). Wavelength scan was performed between 200 nm and 800 nm. Baseline was recorded prior to each set of experiments. Fluorescence measurements were carried out on a Varian Cary Eclipse fluorescence spectrophotometer. Emission spectra were recorded from 380 nm to 800 nm with excitation wavelength at 410 nm, at 25 °C using 1 cm path quartz cells. Relative fluorescence quantum yields were determined according to Miller using Equation (3): (3)$$\Phi_{x}=\Phi_{x}\times A_{s}D_{x}n_{x}^{2}/A_{x}D_{s}n_{s}^{2}$$
where *Φ* is the emission quantum yield; A is the absorbance at the excitation wavelength; *D* is the area under the corrected emission curve and n is the refractive index of the solvents used. The subscripts s and x refer to the standard and to the unknown, respectively. The standard employed was quinine sulphate with a published fluorescence quantum yield of 0.54 [71]. 

### 3.2. UV-Vis Absorbance and Fluorescence Spectra Measurements 

Stock solutions of compounds were prepared by dissolving 1.0 mg of compound in 100.0 mL methanol to form a 1.97 × 10^−5^ M **(NI-1**) and 1.96 × 10^−5^ M (**NI-2**) solutions.

#### 3.2.1. The Effect of pH 

In order to study the effect of pH on the spectroscopic properties of new compounds, UV-Vis and fluorescence emission spectra were recorded covering the pH range 1.6–8.0 for **NI-1** or 1.2–8.0 for **NI-2**. Testing solutions of compounds were prepared by diluting 1.0 mL of stock solution with 2.0 mL buffer and the final concentrations of **NI-1** and **NI-2** for the absorbance and fluorescence measurements were 6.56 × 10^−6^ M and 6.54 × 10^−6^ M, respectively. 

#### 3.2.2. Metal Ion Sensing

All measurements of fluorescent properties were obtained in MeOH/buffer pH 5.5 solution (1:2) with excitation wavelength at 410 nm. The visual evaluation of **NI-1** and **NI-2** sensing properties towards metal cations were performed prior to spectroscopic measurements. The experiment was carried out in 1.0 mL MeOH/buffer mixture (1:2) at a compound concentration 6.56 × 10^−6^ M and 6.54 × 10^−6^ M, respectively, followed by addition of 30.0 µL (2.00 equiv.) of each metal ion stock solution. 

The spectroscopic evaluation of the metal-sensing properties of **NI-1** was carried out at a concentration of 5.0 × 10^−7^ M. The effect of metal cation on emission spectra was tested by adding 3.0, 30.0 and 150.0 µL of stock solution of each metal cation. For titration experiments, in the prepared solution of compound, small aliquots of stock solutions of metal ions were added. The resulting solution of compound and metal ions were mixed and the fluorescence spectra were recorded after 2 min. 

For the evaluation of the sensing properties of **NI-2** towards different metal cations, spectroscopic measurements were obtained at a compound concentration of 6.54 × 10^−6^ M, followed by addition of 150.0 µL of metal ion stock solution. 

## 4. Conclusions

In this paper we investigated the photophysical characteristics of two novel 1,8-naphthalimide-based fluorescent compounds, **NI-1** and **NI-2,** and their potential use as optical probes for pH and heavy metal ions. Different molecular moieties attached at the C-4 position in **NI-1** and **NI-2** show a notable effect on the optical properties of the derivatives. **NI-1** bearing a methyl-pyridyl substituent exhibits 6-fold higher quantum yield than hydroxyphenyl substituted **NI-2**. Both **NI-1** and **NI-2** showed fluorescence intensity increase with decreasing pH. The corresponding p*K*_a_ values for **NI-1** and **NI-2** were determined to be 4.98 and 2.91, respectively. 

When it comes to metal sensing, **NI-1** and **NI-2** show opposite behaviors. The compound **NI-1** shows high sensitivity and selectivity toward Cu^2+^ and Hg^2+^, via fluorescence quenching (switch “off” probe), while the compound **NI-2** undergoes significant fluorescence enhancement exclusively by Hg^2+^ ions (switch “on” probe).

The fluorescence intensity of **NI-1** decreases upon complexation with Cu^2+^ and the chemosensor can be applied to the quantification of Cu^2+^ ions within a linear range from 2.5 × 10^−7^ M to 4.0 × 10^−6^ M with a detection limit of 1.5 × 10^−8^ M. The linear range for Hg^2+^ was determined to be 2.0 × 10^−6^ M to 1.0 × 10^−5^ M, with a detection limit of 8.8 × 10^−8^ M. 

Coordination of the receptor in **NI-2** with Hg^2+^ hinders the PET mechanism from the aliphatic amino group to the naphthalimide core, resulting in fluorescence enhancement. Therefore, the **NI-2** compound can be utilized as a turn-on PET sensor for Hg^2+^ ions. The results show a remarkable selectivity towards Hg^2+^ ions, with a linear detection range from 1.3 × 10^−6^ to 2.5 × 10^−5^ M and a limit of detection of 4.1 × 10^−7^ M. 

Due to the presence of a 2-hydroxyethylsulfonyl group, the covalent immobilization of these naphthalimide probes to different substrate materials and to nanoparticles (e.g., cellulose, textiles) is possible, opening up a wide range of solid state applications for these molecules in the future. 

## Figures and Tables

**Figure 1 molecules-28-01275-f001:**
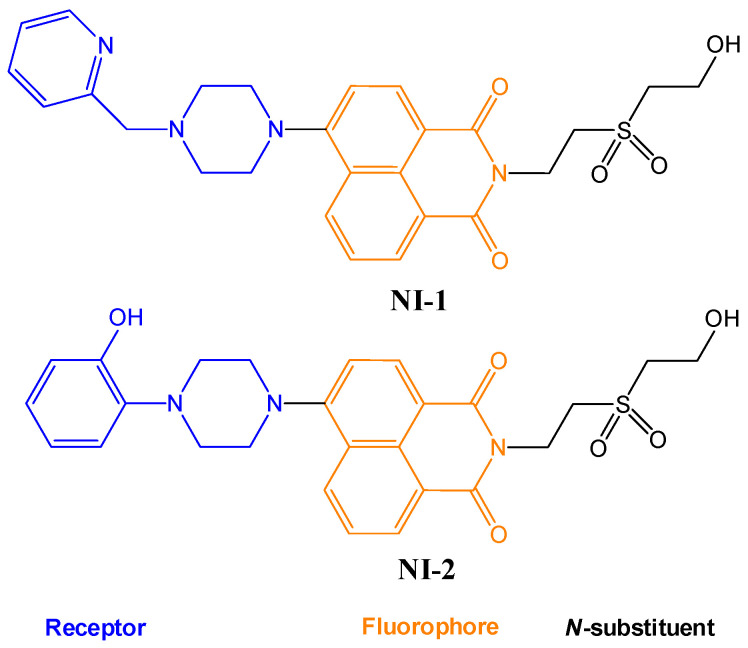
Structures of **NI-1** and **NI-2**.

**Figure 2 molecules-28-01275-f002:**
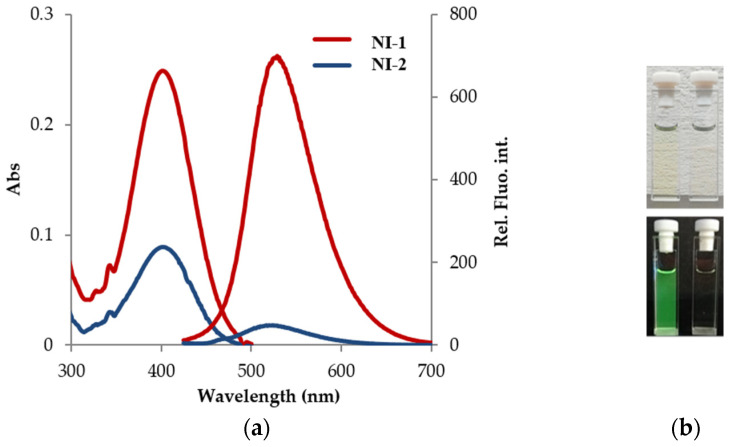
(**a**) UV-Vis absorption and fluorescence emission spectra of **NI-1** and **NI-2** solutions in methanol (*c* = 1.0 × 10^−5^ M, *λ*_exc_ = 410 nm); (**b**) Images of solutions in daylight (above) and under UV lamp (365 nm) (down).

**Figure 3 molecules-28-01275-f003:**
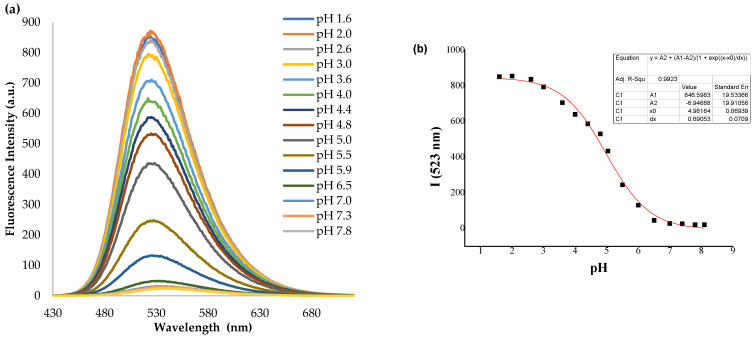
(**a**) Fluorescence spectra of **NI-1** in buffer methanol mixture (2:1) at different pH; (**b**) fluorescence intensity at 523 nm vs. pH.

**Figure 4 molecules-28-01275-f004:**
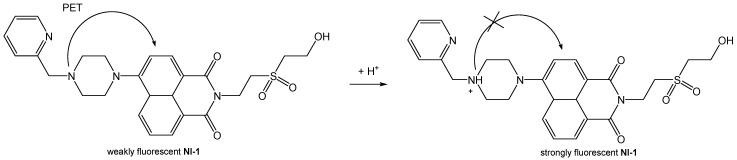
Schematic representation of PET process using **NI-1** as an example.

**Figure 5 molecules-28-01275-f005:**
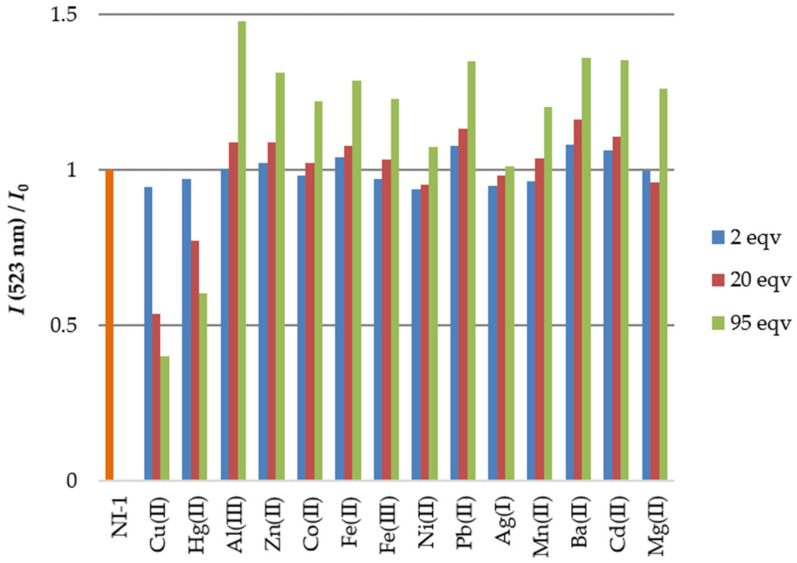
Change of fluorescence intensity at 523 nm for **NI-1** (orange) and in the presence of metal ions in MeOH/buffer (*v*/*v*, 1/2, pH = 5.5) solution.

**Figure 6 molecules-28-01275-f006:**
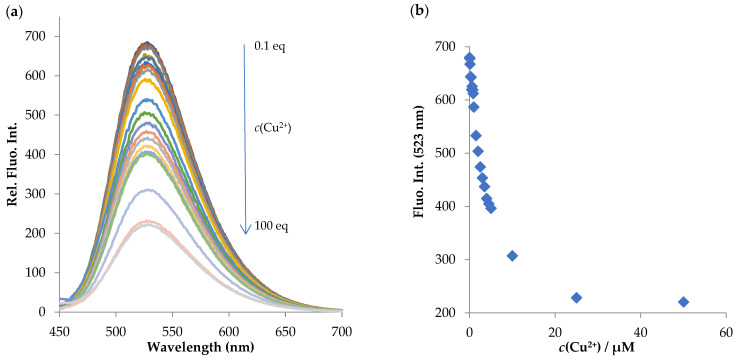
(**a**) Fluorescence titration spectra of **NI-1** (*c* = 5.00 × 10^−7^ M) in the presence of changing concentration of Cu^2+^ ions (*c*(Cu^2+^) = 5.00 × 10^−8^ M–5.00 × 10^−5^ M, as shown in the graph ranging from 0.1 eq to 100 eq) in MeOH/buffer (*v*/*v*, 1/2, pH = 5.5), *λ*_exc_ = 410 nm. (**b**) Plot of fluorescence quenching at 523 nm of **NI-1** as a function of Cu^2+^ ion concentration.

**Figure 7 molecules-28-01275-f007:**
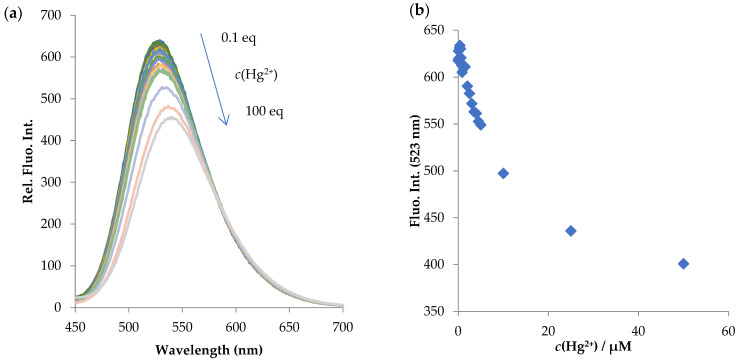
(**a**) Fluorescence titration spectra of **NI-1** (*c* = 5.00 × 10^−7^ M) with Hg^2+^ in concentration range 5.00 × 10^−8^ M–5.00 × 10^−5^ M (shown in the graph ranging from 0.1 eq to 100 eq) in MeOH/buffer (*v*/*v*, 1/2, pH = 5.5), *λ*_exc_ = 410 nm. (**b**) Plot of fluorescence quenching at 523 nm of **NI-1** as a function of Hg^2+^ ion concentration.

**Figure 8 molecules-28-01275-f008:**
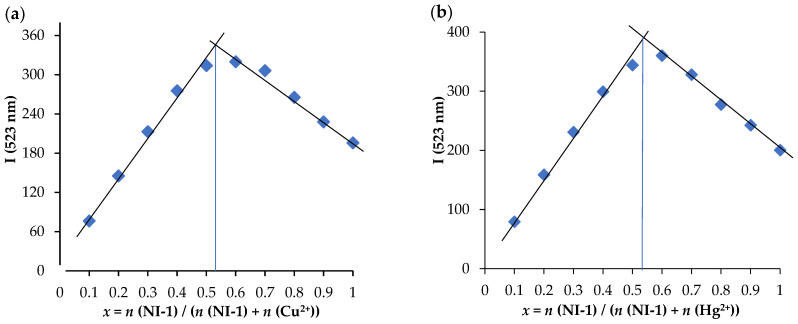
Dependence of fluorescence intensity at 523 nm on the molar content of **NI-1** in solution with (**a**) Cu^2+^ ions and (**b**) Hg^2+^ ions at a total concentration of 5.00 × 10^−6^ M and addition of buffer pH 5.5. The extrapolated directions of intensity dependence on the molar fraction of **NI-1** are presented.

**Figure 9 molecules-28-01275-f009:**
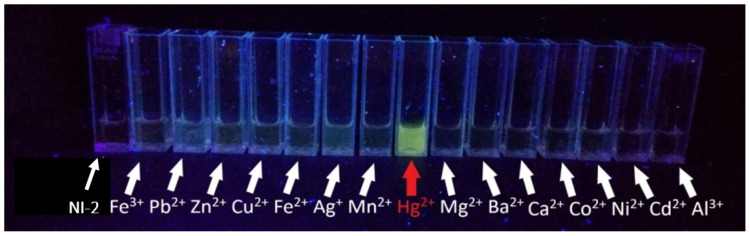
**NI-2** in the presence of metal ions in MeOH/buffer mixture.

**Figure 10 molecules-28-01275-f010:**
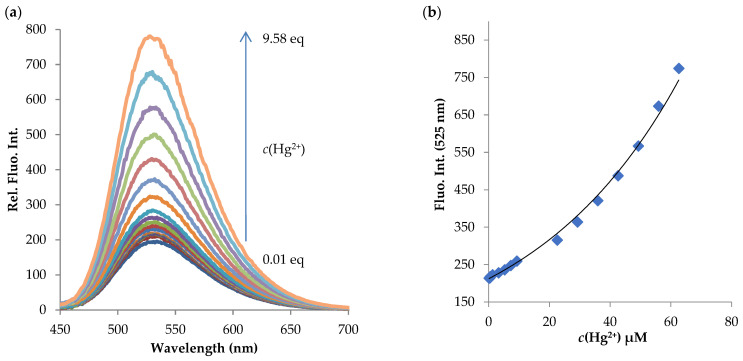
(**a**) Fluorescence titration spectra of **NI-2** (5.00 × 10^−7^ M) with Hg^2+^ in concentration range 1.31 × 10^−7^ M–2.26 × 10^−5^ M (shown in the graph ranging from 0.01 eq to 9.58 eq) in MeOH/buffer (1/2, *v*/*v*, pH = 5.5), *λ*_exc_ = 410 nm. (**b**) Plot of fluorescence enhancement at 525 nm of **NI-2** as a function of Hg^2+^ ion concentration.

**Figure 11 molecules-28-01275-f011:**
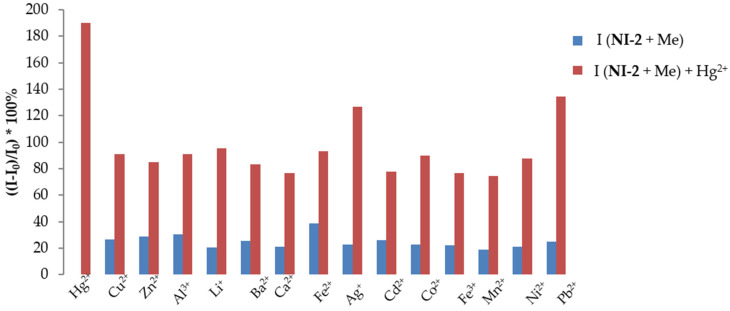
Relative fluorescence intensity changes at 525 nm of **NI-2** (6.54 × 10^−6^ M) in MeOH/buffer solution (*v*/*v*, 1/2, pH 5.5) after addition of 150 µL stock solution of Hg^2+^ and other metal ions after 5 min from the addition of Hg^2+^ ions. The concentrations of Hg^2+^ ions and competing metal ions are equal (4.55 × 10^−5^ M).

**Table 1 molecules-28-01275-t001:** Effect of pH on spectral properties of **NI-1** and **NI-2** in neutral and protonated/cationic forms at pH 1.6 and pH 7.0, respectively.

	pH (Form)	*λ*_abs_/nm	*ε*/M^−1^ cm^−1^	*λ*_emiss_/nm	Rel. Fluo. Int.	Stokes Shift/cm^−1^	p*K*_a,abs_	p*K*_a,emiss_
NI-1	1.6 (NI-1H^+^)	388	2662.90	523	836.2	6653	5.09	4.98
	7.0 (NI-1)	410	2020.58	531	30.2	5558		
NI-2	1.6 (NI-2H^+^)	388	988.53	525	769.8	6726	3.18	2.91
	7.0 (NI-2)	410	916,58	533	<4.95	5629		

**Table 2 molecules-28-01275-t002:** Solvent systems, fluorescence changes, mechanism, stoichiometry (Ligand: Analyte), *K* and LOD values of sensors on complexation with Cu^2+^ ions.

Sensor Structure	Solvent Systems	Fluorescence Changes and Mechanism	Stoichiometry	*K*/M^−1^	LOD/M	Ref.
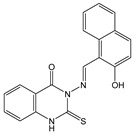	DMSO: H_2_O (1:9, *v*/*v*) at pH = 7.4 (HEPES)	Turn-off at 434 and 494 nm, Paramagnetic quenching	1:1	-	-	[56]
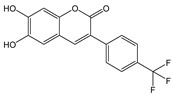	CH_3_CN: HEPES (95:5, *v*/*v*)	Turn-off at 458 nm, Paramagnetic quenching	1:1	9.0 × 10^4^ (BH)	2.45 × 10^−8^	[57]
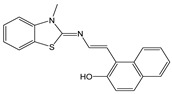	CH_3_CN	Fluorescence enhancement at 540 nm (turn-on), CHEF and ICT	1:1	2.0 × 10^3^ (BH)	3.3 × 10^−6^	[58]
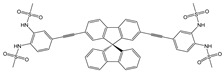	CH_3_CN: HEPES (1:1, *v*/*v*) pH = 7.0	Quenching at 530 nm (turn-off)	1:1	-	9.82 × 10^−8^	[59]
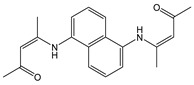	H_2_O: DMF (9.9:0.1, *v*/*v*)	Turn-off at 410 nm, MLCT and paramagnetic quenching	1:2	9.23 × 10^8^ (SV)	1 × 10^−7^	[60]
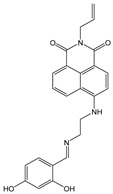	DMF: Tris-HCl buffer (1:1, *v*/*v*)	Turn-off at 528 nm, Paramagnetic quenching	2:1	4 × 10^12^	1.92 × 10^−7^	[50]
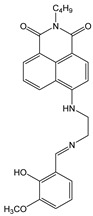	MeOH: HEPES (1:1, *v*/*v*)	Turn-off at 525 nm, Paramagnetic quenching	2:1	1.1 × 10^4^	5.67 × 10^−7^	[53]
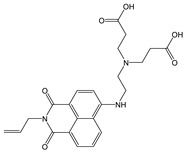	DMSO: HEPES (1:1, *v*/*v*)	Turn-off at 523 nm	1:1	1.14 × 10^6^	4.67 × 10^−8^	[54]
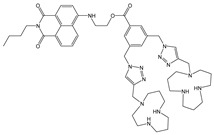	Tris-HCl buffer	Turn-off at 541 nm	1:2	n.d.	2.62 × 10^−9^	[39]
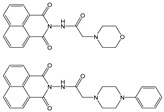	HEPES	Turn-off at 396 nm, Paramagnetic quenching	1:1 1:1	5.7 × 10^5^ 2.5 × 10^5^	1.5 × 10^−6^ 2.5 × 10^−6^	[26]
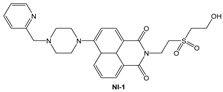	MeOH: phosphate buffer (1:2, *v*/*v*)	Turn-off at 523 nm	1:1	3.6 × 10^5^ (BH) 1.5 × 10^5^ (SV)	1.5 × 10^−8^	This work

*K* values were determined by using Benesi–Hildebrand (BH) or Stern–Volmer (SV) graphical methods.

**Table 3 molecules-28-01275-t003:** Solvent systems, fluorescence changes, mechanism, stoichiometry (Ligand: Analyte), *K* and LOD values of sensors on complexation with Hg^2+^ ions.

Sensor Structure	Solvent Systems	Fluorescence Changes And Mechanism	Stoichiometry	*K*/M^−1^	LOD/M	Ref.
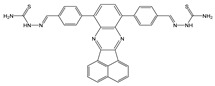	DMSO: H_2_O (1:9, *v*/*v*)	Quenching at 510 nm and red shift (50 nm), AIEE	1:2	2.51 × 10^5^ (BH)	9.07 × 10^−7^	[64]
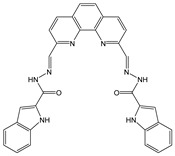	H_2_O with 0.5 % DMSO	Quenching at 480 and 540 nm	1:2	1.77 × 10^5^ (SV)	21.72 ppb	[65]
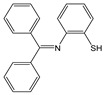	CH_3_CN: H_2_O (1:1, *v*/*v*)	Turn-on at 415 nm and new band at 473 nm, C=N isomerization and ESIPT inhibition	2:1	4.484 × 10^5^ (BH)	2.27 × 10^−8^	[66]
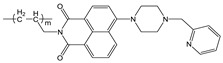	Tris-HCl buffer	Turn-on t at 520 nm, PET inhibition	1:1	3.5 × 10^3^ (BH)	2.0 × 10^−6^	[37]
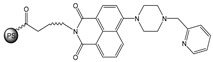	CH_3_CN: HEPES (1:1, *v*/*v*)	Enhancement 405 nm, PET inhibition	-	-	1.98 × 10^−6^	[49]
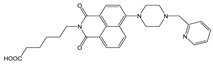	Tris-HNO_3_ buffer	Enhancement at 529 nm, PET inhibition	1:1	2.08 × 10^5^ (BH)	4.93 × 10^−8^	[36]
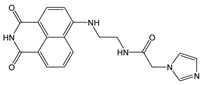	Phosphate buffer	Quenching at 550 nm	1:1	-	2.1 × 10^−6^	[63]
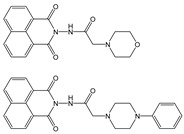	HEPES	Quenching at 396 nm, PET inhibition	1:1	1.8 × 10^5^ 2.1 × 10^5^ (SV)	1.9 × 10^−6^ 2.3 × 10^−6^	[26]
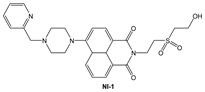	MeOH: phosphate buffer (1:2, *v*/*v*)	Quenching at 523 nm and red shift (18 nm)	1:1	3.9 × 10^4^ (BH) 2.4 × 10^4^ (SV)	8.8 × 10^−8^	This work
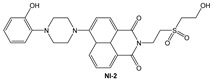	MeOH: phosphate buffer (1:2, *v*/*v*)	Enhancement at 525 nm	-	8278 (BH)	4.1 × 10^−7^	This work

*K* values were determined by using Benesi–Hildebrand (BH) or Stern–Volmer (SV) graphical method.

## Data Availability

The data presented in this study are available in this article and the Appendix A.

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
