# Peer review of "Naphthalimide-Piperazine Derivatives as Multifunctional “On” and “Off” Fluorescent Switches for pH, Hg2+ and Cu2+ Ions"

_molecules, 2023, doi:10.3390/molecules28031275_

Round 1

Reviewer 1 Report

The paper "Naphthalimide-piperazine derivatives as multifunctional “on” and “off” fluorescent switches for pH, Hg2+ and Cu2+ ions" is devoted to the study of photophysical properties of two naphthalimide-piperazine based indicators. They were applied for pH control and revelation of mercury(II) and copper(II) ions in the solution. The protonation and stability constants as well as quantum yields and LODs were calculated and clearly presented. I liked a lot the first part of the paper, which is devoted to the pH-sensing ability of the ligands as there are presented a thorough approach to the mechanism of quenching and a rock-solid method of pKa calculation. Unfortunately, the sections describing the interaction with metals are a bit weaker, and I below will try to explain why I think so.

Major points (or rather one big point):

Lines 323-327. You write about the binding site of mercury(II) ion by ligand, and the reader can understand there that there is the only one site for Hg2+, because there is the only substituent. However, later, on lines 350-353 one can read that the stoichiometry Hg:L = 2:1. Where goes the second mercury(II) ion?

In my opinion, you are correct on lines 323-327, and Job's plot failed to reveal the real stoichiometry of the complex. It is not Authors' fault; it is an intrinsic feat of Job's method (see papers 10.1039/C6CC03888C, 10.1021/acs.joc.5b02909 for details).

Furthermore, I also doubt in values of stability constants determined by Benesi-Hildebrand method (see paper 10.1016/j.saa.2020.119334 for details) unlike pKa, which were calculated from fitting procedure of titration curve. Stability constants should be derived from the experimental titration curves by similar procedure using suitable software (e.g. k-ev.org/).

I also did not like the way the sections "Reversibility tests" are currently made. In my opinion, it would be better to give the speciation diagrams (instead of Fig. S7) calculated for quaternary systems Ligand:Cu:EDTA:Hg or, at least just list the reactions (as well as their constants) which occurs in such systems viz. 1) ligand complexation with Cu2+; 2) ligand complexation with Hg2+; 3) EDTA complexation with Cu2+; 4) EDTA complexation with Hg2+. By the way, I still agree with Authors (lines 267-269) that Hg2+ will replace Cu2+ in complexes, if Hg2+, Cu2+, EDTA and ligand coexist because EDTA binds copper(II) far more strongly than mercury(II). It should be just presented more clearly.

By the way, If I understood correctly, the metal ions interact with the protonated indicator molecules, do not they? To make sure, ideally, the protolytic equilibria should be considered as well when the complex formation constants are calculated.

Minor points:

1. Why do you believe that proton is accepted by this nitrogen of piperazine moiety, but not another?

2. I believe you are aware that pH value reported for the binary solvents of  methanol/aqueous buffer refers to the aqueous buffer, but not the binary mixture. It is an important nuance, which should be underlined in the text.

3. Ideally, the error bars should be added to all the histograms to help readers to understand how selective the indicator actually are. Without error bars, Fig. 11, actually, do not impress. the same applies to the reversibility charts; for example, looking upon Fig. S5, one could think that indicator do not only worsen with every cycle of using - it actually becomes better and better! It is hardly so, isn't it? The reason of increase in fluorescence intensity is perhaps just an error of its measurements.

4. However, if the error is too large, it questions the standard deviation of blank measurements used for LOD calculation. SD of blank should somehow correspond to the SD value elsewhere; or the differences should be explained.

5. It is well-known that Stern-Volmer constant is actually a product of the bimolecular quenching rate constant, and the fluorescence lifetime of the unquenched fluorophore (10.1016/j.molstruc.2011.05.023). Therefore, it makes sence to determine the lifetime to confirm your guesses (I find them plausible, but nonetheless). Might it be, for example, that the real mechanism of quenching is just collisional one?

6. Finally, it is difficult for readers to evaluate the worthiness of newly reported indicators without their comparison with the known analogs or other methods of metal concentration determination. Therefore, a short table with several known from the literature methods is probably in order (e.g., for Cu2+: 10.1016/j.jphotochem.2014.02.015, 10.1007/s10895-020-02503-4, 10.1007/s10895-021-02752-x, 10.1021/acsomega.1c02744, 10.1021/acsomega.8b00748, 10.3390/inorganics10070102, 10.7324/JAPS.2018.8206; for Hg2+: 10.1016/j.snb.2016.06.129, 10.1016/j.ica.2017.09.040, 10.1016/j.tetlet.2016.08.031).

It is a very good paper although there is some romm for improvement.

Reviewer 2 Report

The manuscript Naphthalimide-piperazine derivatives as multifunctional “on” and “off” fluorescent switches for pH, Hg2+ and Cu2+ ions, is well described and is experimentally well performed, however the results presented are not sufficiently relevant and innovative. I recommend the following modifications to improve its quality.

1. The authors describe the fragments attached to the piperazine ring (pyridin-2-ylmethyl and phenol) as functional group, I propose to remove this term and replace it by scaffold, chemical strucutre...…

2. The authors explain that NI-1, bearing pyridine functionality is strongly green fluorescence emitting fluorophore, compared to hydroxyphenyl-substituted NI-2, by an acceptor effect of the pyridine ring, however, different to hydroxyphenyl, it is not directly bound to the piperazine nitrogen, therefore, the discussion of these results should be performed with a new perspective.

3. Line 104, the authors said "where pH is higher than 7.4, a decrease of the peak intensity occurs", however this effect is not observed in the S1a figure.

4. PH effect on Naphthalimide-piperazine skeleton fluorescence is well known and has been widely reported. I suggest include more reference.

Fu, Y., Zhang, J., Wang, H., Chen, J. L., Zhao, P., Chen, G. R., & He, X. P. (2016). Intracellular pH sensing and targeted imaging of lysosome by a galactosyl naphthalimide-piperazine probe. Dyes and Pigments,133, 372-379.

Lee MH. Bis(Naphthalimide-Piperazine)-Based Off-On Fluorescent Probe for Acids. J Fluoresc. 2016 May;26(3):807-11.

5. The authors propose the protonation of NI-2 at the nitrogen of position 4 of the piperazine ring, although this protonation is evident and widely reported for the 4N alkyl derivatives, in contrast, it is not so evident for NI-2, since the two nitrogens of the piperazine ring are linked to aryl moieties. Possibly, the difference observed between NI-1 and NI-2 (29.6-fold for NI-1 and 162.5-fold for NI-2), is based on this difference. I suggest to the authors to conduct theoretical studies or literature data to support the proposed mechanism for NI2.

6. The authors perform the spectral properties measurements in buffer pH 1.6 and 7, however the metal ion sensing tests were performed in MeOH/buffer (v/v, 1/2, pH 5.5). The authors should perform the study of the properties under the same conditions in which the subsequent tests are carried out.

7. Authors should report the salt used for the analysis of the different metal cations.

8. The fluorescent probes described by the authors are quite sensitive, particularly to develop turn-off sensors, and the sensitivity should be increased. It is unclear why the authors have chosen measurements of cation interactions at a pH of 5.5.

9. The paragraph on lines 229-243 is unclear and I suggest that it be reviewed.

Round 2

Reviewer 1 Report

Authors made the necessary improvements, gave answers to the concerns raised in satisfactory manner, and it is my pleasure to recommend the publication of the manuscript in the revised form.

Reviewer 2 Report

The authors have improved the quality and clarity of the manuscript Naphthalimide-piperazine derivatives as multifunctional “on” and “off” fluorescent switches for pH, Hg2+ and Cu2+ ions. I consider it ready for publication. If possible, improve the quality of figure 1a.